# Adaptation and Formative Evaluation of Online Decision Support to Implement Evidence-Based Strategies to Increase HPV Vaccination Rates in Pediatric Clinics

**DOI:** 10.3390/vaccines11071270

**Published:** 2023-07-21

**Authors:** Ross Shegog, Lara S. Savas, Erica L. Frost, Laura C. Thormaehlen, Travis Teague, Jack Steffy, Catherine Mary Healy, Laura Aubree Shay, Sharice Preston, Sally W. Vernon

**Affiliations:** 1Department of Health Promotion and Behavioral Sciences, University of Texas School of Public Health, Houston, TX 77030, USA; lara.staub@uth.tmc.edu (L.S.S.); erica.l.frost@uth.tmc.edu (E.L.F.); laura.c.thormaehlen@uth.tmc.edu (L.C.T.); travis.a.teague@uth.tmc.edu (T.T.); laura.aubree.shay@uth.tmc.edu (L.A.S.); sharice.m.preston@uth.tmc.edu (S.P.); sally.w.vernon@uth.tmc.edu (S.W.V.); 2Department of Anthropology, Washington University in St. Louis, St. Louis, MO 63130, USA; j.steffy@wustl.edu; 3Department of Pediatrics, Infectious Diseases Section, Baylor College of Medicine, Houston, TX 77030, USA; chealy@bcm.edu

**Keywords:** HPV vaccination, pediatrics, decision support, implementation, digital health, usability, feasibility

## Abstract

Human papilloma virus (HPV) vaccination rates remain below national goals in the United States despite the availability of evidence-based strategies to increase rates. The Adolescent Vaccination Program (AVP) is a multi-component intervention demonstrated to increase HPV vaccination rates in pediatric clinics through the implementation of six evidence-based strategies. The purpose of this study, conducted in Houston, Texas, from 2019–2021, was to adapt the AVP into an online decision support implementation tool for standalone use and to evaluate its feasibility for use in community clinics. Phase 1 (Adaptation) comprised clinic interviews (*n* = 23), literature review, Adolescent Vaccination Program Implementation Tool (AVP-IT) design documentation, and AVP-IT development. Phase 2 (Evaluation) comprised usability testing with healthcare providers (HCPs) (*n* = 5) and feasibility testing in community-based clinics (*n* = 2). AVP-IT decision support provides an Action Plan with tailored guidance on implementing six evidence-based strategies (immunization champions, assessment and feedback, continuing education, provider prompts, parent reminders, and parent education). HCPs rated the AVP-IT as acceptable, credible, easy, helpful, impactful, and appealing (≥80% agreement). They rated AVP-IT supported implementation as easier and more effective compared to usual practice (*p* ≤ 0.05). The clinic-based AVP-IT uses facilitated strategy implementation by 3-month follow-up. The AVP-IT promises accessible, utilitarian, and scalable decision support on strategies to increase HPV vaccination rates in pediatric clinic settings. Further feasibility and efficacy testing is indicated.

## 1. Introduction

In the U.S., infections of human papillomavirus (HPV) approximate 79 million people, with an expected 14 million newly infected persons each year [1]. Persistent infection with high-risk HPV types (mainly 16 and 18) are associated with cervical and anal cancers (>90%), oropharyngeal cancers (70%), vaginal and vulvar cancers (70%), and penile cancers (>60%) and with associated increases in health costs [2,3,4,5,6,7,8]. The HPV vaccine is efficacious in decreasing HPV infections, precancerous lesions, and genital warts [9,10,11,12]. The Advisory Committee on Immunization Practices (ACIP) recommends the HPV vaccination series be initiated in both males and females by 11 to 12 years of age [13,14]. There is evidence for the safety [15,16] and effectiveness [10] of HPV vaccination, yet rates are below the Healthy People 2030 goal of 80% for series completion in adolescents 13 to 15 years of age [17]. Rates are also below those of acellular pertussis and meningococcal conjugate vaccines [18]. A robust evidence base exists for strategies that increase HPV vaccination, including healthcare provider (HCP) assessment and feedback [19], prompts to providers [20], patient reminder systems [21], and both provider- and patient-directed education when implemented in combination with other healthcare-system-based strategies [22]. Healthcare-system-based implementation strategies are effective because they can operate simultaneously at the organizational, provider, and patient levels. Despite this, there is inconsistency in the strategies that are adopted and the degree of implementation across clinics, even within clinical networks [22,23,24].

The Adolescent Vaccination Program (AVP) is a multi-component program to increase HPV vaccination rates in pediatric clinics. The AVP targets HCPs and parents of pediatric patients 11–17 years of age [23]. The AVP contains a suite of evidence-based strategies that include: (1) immunization champions, (2) provider assessment and feedback, (3) provider continuing education, (4) provider prompts, (5) parent (patient) reminders, and (6) parent (patient) education. The AVP has been successful in significantly increasing initiation and completion of the HPV vaccine series in large urban networks in Houston and San Antonio, Texas [24,25,26].

The existence of such evidence-based strategies does not mean that clinics will necessarily adopt or implement them. Therefore, implementation researchers have compiled a set of effective implementation strategies identified to support implementation of evidence-based clinical practices [27]. Delivery of implementation strategies using theoretically and empirically based decision support tools could assist HCPs and clinic managers to determine their clinic’s readiness to adopt evidence-based strategies and could provide utilitarian guides for their implementation.

Online decision support tools can effectively aid decision making in varied settings, including healthcare [28]. They can enhance HCP practice in drug dosing, preventive care, and active medical care [29,30]. Given their effectiveness, experts in dissemination research have encouraged the use of decision support tools to translate evidence-based science into practice [31]. 

The efficacy of the AVP and the potential of online decision support in enhancing quality improvement in clinical settings suggest the potential benefits of adapting the AVP to be an online decision support tool. This implementation support tool could enable clinics to independently implement the bundle of six evidence-based strategies under their own auspices and provide scalability to the AVP to accommodate broad geographic reach. 

Individual and organizational factors can facilitate decisions to adopt and implement technology-based interventions [32]. These factors include individual user perceptions of the intervention’s acceptability, credibility, ease of use, simplicity, scope, benefit, impact, understandability, duration, scalability, relative advantage, and appeal [33]. They also include an organization’s capacity to support implementation that include management support, resources, and accommodation in existing schedules and flow [34]. Usability and feasibility testing of intervention prototypes is an important formative step to ensure confidence that the digital intervention functions the way it is designed to while not disrupting or comprising the organizational flow [35].

The purpose of this study was to (1) adapt the AVP to provide online decision support to implement six evidence-based AVP strategies in pediatric clinics throughout Texas, (2) assess its usability and acceptability with end-users, and (3) assess the feasibility for its use in community clinic settings. This study contributes to the literature on adapting existing evidence-based protocols into online decision support applications.

## 2. Materials and Methods

The Adolescent Vaccination Program Implementation Tool (AVP-IT) is a decision-support website designed to support the implementation of AVP strategies into pediatric clinics. The development of the AVP-IT comprised 2 phases to: (1) adapt the existing in-person AVP evidence-based program into a completely Internet-based decision support program for implementation (AVP-IT) and (2) formatively evaluate the AVP-IT for usability and feasibility (Table 1). This study was conducted in Houston, Texas, USA, between March, 2019 and November, 2021.

### 2.1. Phase 1: Adaptation

Goals for adaptation were to ensure that original content, objectives, and functionality of the AVP was retained in the new online program and that it was acceptable to end users, feasible for delivery in the community clinic setting, and potentially impactful in motivating change in clinic practice. Stepped translation frameworks described by Card et al. (2011) and Bartholomew et al. (2006), respectively, guided the incorporation of theoretically and empirically based evidence in the adaptation process [36,37].

#### 2.1.1. Step 1: Semi-Structured Interviews

Active participatory research with an expert advisory group informed the translation of existing AVP protocols throughout the AVP-IT development. The expert advisory group comprised 10 decision-makers and stakeholders representing pediatric clinic management, pediatric clinical healthcare providers (HCPs), researchers in public health immunization, and representatives from community organizations with a mission focus on immunization. The expert advisory group was established to formatively evaluate content, strategies, and methods to include in the AVP-IT and how to best package the tool to support implementation and ensure generalizability to a breadth of clinic environments.

Expert advisory group members assisted in identifying HCPs and clinic managers from a sample of 23 clinics that varied by size (small of <3 HCPs vs. large of ≥3 HCPs), location (rural vs. urban), affiliation (single vs. network affiliated), medical record type (EMR vs. paper-based), and primary patient payment (Medicaid vs. commercial insurance). Clinic heterogeneity was important for generalizability, ensuring the tool has utility in varied clinic environments. An ethnographic protocol was employed to describe facilitators and barriers within Texas clinics to (1) implement evidence-based strategies to increase HPV vaccination rates and (2) adopt and implement an online tool to guide this implementation [38,39,40,41].

HCPs and clinic managers were invited to provide their perspectives in 30 min semi-structured phone interviews on the features of a decision support system that would promote acceptability, utility, and ease of use. Semi-structured interviews were selected (as opposed to structured interviews) because they provided more scope for respondents to elaborate on their perspective while still maintaining enough structure for interviews to be compared [39]. Measurement instruments comprised an interview guide that focused on the barriers and facilitators to implementing online decision support and a brief survey to confirm each clinic’s structure and demographic characteristics. The interview guide was designed to understand the daily functions of the clinics, how changes are implemented, and how a decision tool could make it easier for clinics to adopt evidence-based strategies. Interviews were analyzed thematically for common themes, including facilitators and barriers, around implementation of a decision support tool.

#### 2.1.2. Step 2: Literature Review

A brief scoping review was conducted to append the research team’s existing literature base. The review was designed to identify: (1) behavioral and environmental factors that impact HPV vaccination; (2) behavioral and environmental barriers to the adoption and implementation of evidence-based strategies designed to increase HPV vaccination (including organizational factors); (3) interventions that have successfully mitigated these barriers; and (4) successful theory-based and empirically based frameworks (e.g., IM-ADAPT, RE-AIM, CFIR, and Fishbein’s Integrative Model of Behavior Change) to enhance adoption and implementation. The research team identified relevant studies and synthesized the literature into evidence tables for review by the expert advisory group. Results are described in Section 3.1.2 below and informed the concept, content, architecture, and functional design specifications of the AVP-IT.

#### 2.1.3. Step 3: Develop the AVP-IT Design Document

The AVP-IT design document was developed to describe program specifications including core content, scope, function, flow, and methods and strategies. Content analysis of the original AVP design document (describing its aims, target population, objectives, behavioral matrices, content, and tools) provided the foundation for AVP-IT design. The Intervention Mapping (IM) framework had guided the original AVP development [23]. Core content, scope, and best practice characteristics were represented in behavioral change matrices, a product of the IM process. These cross-referenced targeted behavioral outcomes, performance objectives, behavioral mediators, and learning objectives [36]. Targeted behavioral outcomes for the original AVP program are described elsewhere [23]. These included: (1) collaboration with AVP champions in each clinic regarding implementation of strategies and immunization rates; (2) HCP review of assessment and feedback reports on their HPV vaccination rates; (3) coordination with clinical staff to provide consistent messaging to patient/parents regarding HPV vaccination; (4) checking for vaccine eligibility at every clinic encounter; (5) delivering strong presumptive recommendations to eligible patients at time of visit; (6) bundling HPV vaccine recommendations with other vaccines; (7) determining patient/parent concerns if they are vaccine hesitant; (8) communicating tailored messages to address specific patient/parent concerns; and (9) reminding patients to schedule follow-up HPV vaccine dose(s) before leaving the clinic.

The original AVP program was designed to target behavioral antecedents of knowledge, self-efficacy, outcome expectations, and normative beliefs. Upon review, the research team determined that the targeted behaviors (performance objectives) and antecedents were relevant and valid for current HPV vaccination behaviors. Design document development was iterative. An expert advisory group reviewed the design for appropriateness, clarity, layout, and language. Theoretical methods (i.e., self-assessment, role modeling, and guided practice) and practical strategies (i.e., stepped Action Plan, role model video, photo-based interactive activities) were included in the AVP-IT design to increase clinic staff knowledge, skills, and self-efficacy. The design document was the ‘blue print’ for the AVP-IT production. Design features for the AVP-IT were informed by the results from Steps 1 and 2 (refer to Section 3.1.3 below for further detail).

#### 2.1.4. Step 4: Produce the AVP-IT

An agile and iterative programming schedule was employed to produce the AVP-IT. The AVP-IT was programmed in HTML5, accessible from any internet-enabled device, and included a data input ‘Wizard’ to create a dynamically built Action Plan. Asset development included role-model testimonial videos, an automated assessment and feedback spreadsheet developed in Excel that enabled conversion of clinic- and provider-based vaccination data (inputted into 10 data cells) into dynamically generated quarterly reports, links to continued medical education (CME) content and previous publications, and embedded tools corresponding to each of the six evidence-based strategies (described more fully in Section 3.1.3 below). A back-end database was produced to track visits and maintain a deidentified record of data responses to data inputted for the Action Plan. The AVP-IT was alpha tested in-house to ensure it conformed to intended design and functional specifications.

### 2.2. Phase 2: Formative Evaluation

Usability and feasibility testing followed previously published protocols from the research team [42].

#### 2.2.1. Step 5: Usability Testing

*Study Design and Participants*: Expert advisory group members (*n* = 5) were sent the AVP-IT website URL, reviewer instructions, and a usability survey in July 2021. They were instructed to access all elements of the AVP-IT—including the Action Plan Wizard, their clinic’s tailored Action Plan, and accompanying resources—and then to complete the usability survey with demographic questionnaire (including items on age, gender, race/ethnicity and job title) and a debriefing session with the research principal investigator. The adapted usability survey assessed functions of the AVP-IT and potential improvements of content, function, and interface design [42]. Participants were asked to pay particular attention to evaluating the content located in the ‘How To’ section of each Action Plan strategy. This was to ensure that user guidance on implementation steps was explained appropriately and was understandable. Usability parameters included acceptability (likability), credibility, ease of use, information scope, duration, helpfulness, perceived impact, and motivational appeal adapted from pre-existing usability assessment instruments [42]. The sample size was suitable for usability testing which is typically descriptive, not requiring statistical significance as a criteria for determining usability problems [43].

*Measurement and Analysis*: *Acceptability* was assessed by how much participants liked the AVP-IT and various components, including the ‘About’ page, Action Plan, and Action Plan Wizard (response ratings: dislike a lot, dislike a little, like a little, like a lot, did not use). *Credibility* was assessed from the perceived correctness of the content (response ratings: accurate, inaccurate, no opinion) and whether the content could be trusted (response ratings: can be trusted, can’t be trusted, no opinion). *Ease of use* was assessed from the perceived difficulty in using the AVP-IT and completing the clinic Action Plan (response ratings: very easy, kind of easy, kind of hard, very hard, did not use). *Complexity* was assessed from perceptions of whether the AVP-IT was unnecessarily complex or cumbersome, if components were well integrated, and if participant’s felt the AVP-IT required technical support or extensive training to be able to use it (response ratings: strongly agree, agree, neither agree nor disagree, disagree, strongly disagree). *Information scope* was assessed by perceptions on the amount of information provided (response ratings: just right, too much, too little). *Duration* was assessed from the perception of how long it would take to use the AVP-IT and complete the Action Plan Wizard (response ratings: too quick, just right, too long). *Helpfulness* was assessed from perceptions of helpfulness of the ‘About’ page (response ratings: helpful, not helpful, don’t know) and the AVP-IT program components (response ratings: extremely helpful, very helpful, somewhat helpful, not at all helpful). *Perceived impact* was assessed from perceptions of whether the AVP-IT would help clinics adopt, implement, and maintain HPV implementation strategies (response ratings: strongly agree, agree, neither agree nor disagree, disagree, strongly disagree, don’t know). *Motivational appeal* was assessed from the likelihood of AVP-IT use by other clinics and if they should use it (response ratings: yes, no, maybe) and the participants’ interest in using the AVP-IT (response ratings: strongly agree, agree, neither agree nor disagree, disagree, strongly disagree). Open-ended questions were included to assess what participants liked best and liked least about the AVP-IT and its three main components: ‘About AVP-IT’, ‘My Action Plan’, ‘My Toolkit.’ Participants were also asked what else they would have liked to see in these components and how the AVP-IT could be made more appealing.

Relative advantage of the AVP-IT was assessed from HCP expectations of impact on implementing evidence-based strategies when using the AVP-IT compared to usual practice (i.e., implementing strategies to increase HPV vaccination without the AVP-IT) using 5-point (range: 1–5) semantic differential scale assessing: ease of use (easier/harder), time (less time/more time), and effectiveness (more effective/less effective) [42]. Each participant was given 10 days to review the AVP-IT website and complete the usability survey. Usability data was assessed using descriptive statistics of central tendency. The semantic differential scale was assessed using a paired 2-tailed *t*-test comparison of responses against neutral ratings using STATA IC Version 17 analytic software [44]. The hypotheses tested was that those exposed to the AVP-IT would perceive its use as providing significant advantage (easier, less time intensive, and more efficacious) compared to their usual practice. Semi-structured 30 min debrief interviews with each participant provided information on recommended enhancements to improve the user experience and to facilitate the adoption and implementation of the AVP-IT in clinics.

#### 2.2.2. Step 6: Feasibility Testing Case Study

A pilot test of the AVP-IT was conducted to determine the feasibility of delivery in typical community clinic settings and its impact. *Study Design and Participants*: A single-group, pre-test/post-test design was conducted in two community clinics, comprising one and four physicians, respectively, that were part of a broader clinic network of seven community clinics in the Greater Houston area. The clinics provided medical, specialty, behavioral health, diagnostic, and immunization services to a patient population comprising mainly Hispanic (43%, 67%), African American (32%, 14%), and South Asian (19%, 11%) patients who were mainly uninsured or self-pay (74%, 94%).

*Measurement and Analysis*: At each site, clinic directors selected an AVP Champion.

AVP Champions could be chief medical officers, practice managers, HCPs, clinic administrators, or clinic staff. They were responsible for coordinating rollout of strategies in accordance with the Action Plan. Champions provided demographic data, including age, gender, education level, race/ethnicity, and job title. They then received an e-mail with the AVP-IT link and further instructions for the feasibility trial. They then visited the online AVP-IT and completed the Action Plan Wizard and printed their clinic’s tailored Action Plan. They were asked to follow the steps in the plan to implement each of the AVP strategies in their clinic and, at 3-month follow-up, to again complete the Action Plan Wizard. The AVP-IT Action Plan was used to assess pre- and 3-month post-test implementation with six items that asked about implementation status (pending, partial, full) based on AVP-IT criteria. Strategies were defined as ‘fully implemented’ if they were implemented in accordance with the AVP-IT Action Plan criteria, or else they were defined as ‘partially implemented’. Strategies defined as ‘pending’ were yet to be initiated. Implementation of AVP strategies was compared descriptively between baseline and at 3-month follow-up. Monitoring call field notes provided qualitative feedback on the AVP-IT implementation process. Over the three-month study period, bimonthly monitoring calls were conducted to obtain status updates and troubleshoot challenges associated with the AVP-IT or implementation of AVP strategies in clinics. The champions were asked to complete 15 min exit interviews with research staff to provide a summary of their experiences and recommendations for AVP-IT enhancements. Champions received a $25 gift card for completing pre- and post-test assessment respectively.

## 3. Results

### 3.1. Phase 1: Adaptation

#### 3.1.1. Step 1. Semi-Structured Interviews

Respondents were clinic staff from five urban clinics comprising two large (≥3 HCPs) and one small clinic (<3 HCPs) with a mainly Medicaid patient population and one large and one small clinic with a mainly commercially insured patient population [41]. Interviews provided information on the varying priorities and preferred strategies for varied clinic types. A clinic’s core mission orientation was a critical factor in adopting and implementing strategies. Clinics with a majority percentage of Medicaid or uninsured patients (e.g., safety net clinics, community-based Federally Qualified Health Centers) and oriented to providing care to a maximum number of underserved patients preferred in-person training for evidence-based strategies to increase HPV rates. They emphasized that strategies of greatest utility were provider prompts, patient reminders, and patient education tools. Clinics with a majority percentage of commercially insured patients (e.g., health maintenance organizations) preferred online training for evidence-based strategies to increase HPV rates. They emphasized that strategies of greatest utility were immunization champions, provider education, and assessment and feedback reports. Structural considerations of clinic size, location, insurance, and EHR were not considered to be critical barriers in the clinic staff’s decision to implement evidence-based strategies to increase HPV vaccination rates or to adopt and implement an online tool to guide this implementation.

#### 3.1.2. Step 2: Literature Review

Tables of facilitators and barriers were constructed to categorize implementation of decision support, multi-component interventions, and evidence-based strategies within the AVP (Table 2 and Table 3). Facilitators to adoption and implementation included the right people receiving the right information at appropriate times, training and leadership support, printed copies of documents and reports, incentivization, adding prompts to visit preparation, and previous messaging experience. Barriers to adopting and implementing strategies within the AVP include lack of funding, staff turnover and lack of staff buy-in, limited access to resources, lack of time and difficulty in changing workflow, and competing priorities.

#### 3.1.3. Steps 3 and 4: AVP-IT Design and Production

The AVP-IT is designed to assist clinic staff in the adoption, implementation, and maintenance of evidence-based HPV vaccination strategies within their clinic. AVP-IT provides guidance in (1) determining the level of readiness of the clinic for implementation of each AVP strategy, (2) implementing each strategy regardless of the size and type of pediatric clinic or clinic network, and (3) increasing systems knowledge and capacity to address low HPV vaccination rates.

For stand-alone functionality, the AVP-IT required a robust, simple, intuitive architecture and navigation and sufficient scaffolding to inform naïve users of its purpose and use. This need prompted core features of (1) a tailored implementation Action Plan dynamically generated in response to data (input into an AVP-IT Action Plan ‘Wizard’) on clinic parameters that provides guidance on rolling out each evidence-based strategy; (2) an automatic assessment and feedback report generator, responsive to each provider’s immunization data that provides graphic confirmation of the provider performance in relation to other providers; and (3) video-based testimonials from credible stakeholders who emphasize the ease and importance of each evidence-based strategy. The AVP-IT website is housed on a secure UTHealth server and is accessible online to any clinic (https://avp.sph.uth.edu/, accessed on 20 May 2023). It comprises four webpages: (1) About AVP-IT, (2) My Action Plan, (3) My Toolkit, and (4) Contact Us, designed for user-friendly and tailored guidance to assist clinics in implementing AVP strategies regardless of clinic size and type (Figure 1).

*AVP-IT use by AVP Champions*. The AVP Champion (previously described) can use the AVP-IT by following the four steps that correspond to the page structure of the AVP-IT site: (1)*About AVP-IT* provides an overview and background on the AVP in video and text with links to research publications.(2)*My Action Plan* provides an Action Plan Wizard. The AVP Champion can enter descriptive data about their clinic (e.g., clinic size, location, EHR platform, affiliation) and status of the six AVP strategies by responding to six multiple choice questions (Table 4). They then receive a printable Action Plan that is dynamically compiled within the AVP-IT based on the answers provided in the Action Plan Wizard. The Action Plan provides tailored feedback responsive to the implementation status (pending, partial, or full) for each strategy, stepped guidance on how to implement each strategy (Table 5), and an array of resources developed from previous AVP studies that are designed to facilitate implementation of each strategy. No sensitive information is collected.(3)*My Toolkit* provides the AVP Champion with an ‘à la carte’ selection of printable resources, video testimonials on the importance of each strategy, and tips on successful implementation.(4)*Contact Us* provides the AVP Champion with a form to contact the AVP research team with questions or to report issues with the AVP-IT website. A back-end database collects metrics including Action Plan data and Google analytic data (e.g., time spent on the AVP-IT and pages and resources accessed).

*Action Plan Printout*: The Action Plan comprises sections that describe each of the six intervention strategies. These are listed in order of the recommended implementation sequence: establish an AVP Champion, link provider to CME, institute quarterly assessment and feedback reports, develop a patient reminder system, develop a provider EMR prompt to cue to the presence of vaccine eligible youth, and promote a patient education app (*HPVcancerFree*) [23,25]. Each of the six strategy sections provides information describing the strategy and its importance: ‘How To’ steps to follow to implement the strategy, ‘Tools’ that link to resources needed in the implementation process, ‘Tips for Success’ that list bulleted practical recommendations for success informed by previous studies, and ‘Quotes from the Field’ that provides tips from HPCs who have implemented the strategies. Downloadable, printable, and/or viewable ‘tools’ include instructional videos, assessment and feedback report templates, algorithms for programming EHR-based provider and patient reminders, promotional flyers (i.e., for the AVP continued medical (CME) and nursing (CNE) education activity and *HPVcancerFree* app), tracking forms, and scripts to facilitate communication. The AVP Champions were responsible for approximately 16 itemized tasks (Table 5).

### 3.2. Phase 2: Formative Evaluation

#### 3.2.1. Step 5: AVP-IT Usability Results

Five members of the expert advisory group completed the usability rating scales. The sample comprised physicians (*n* = 3), a clinic administrator (*n* = 1), and a community non-profit manger (*n* = 1), who were mostly male, 50–59 years of age, and non-Hispanic white (Table 6). Most participants reported reviewing all the AVP-IT pages: About AVP-IT’ (5 of 5), Action Plan Wizard (3 of 5), tailored Action Plan printout (3 of 5), and ‘My Toolkit’ (5 of 5). The AVP-IT was rated positively for most usability parameters assessing acceptability (Table 7) and utility (Table 8). Three participants completed a 30 min follow-up debriefing session.

*Acceptability:* All participants rated the AVP-IT and Action Plan as acceptable. *Credibility:* The information in the AVP-IT was rated as accurate and trustworthy by all respondents. *Ease of use:* All participants rated the AVP-IT was easy to use, stating that it is “easy to use and easy to navigate” (Participant 3) and that “It can be a good tool for clinics who have a state of readiness so they can develop a strategy … it was easy to utilize.” (Participant 4). *Complexity:* All participants that the AVP-IT functions were well integrated. At least four of five participants agreed that the AVP-IT was not overly complex, that the AVP-IT did not require technical support to be able to use it, and that extensive learning was not necessary for its use. Participants stated that they liked the “organization” (Participant 1), “the sequence and flow of the overview [and] the simplicity” (Participant 4), and that it was “clean and clear” (Participant 3). They noted that “the Action Plan was very detailed” (Participant 1) and liked the “option to review information in multiple ways (pop-ups, downloads)” on the tools page (Participant 4). *Information scope:* All participants rated the AVP-IT and the tailored Action Plan as providing an appropriate amount of information stating that it “has good information … it sums up and gives clear details regarding the work to improve HPV vaccination rates.” (Participant 3) *Duration:* Participants stated that AVP-IT was “short and snappy, I didn’t have to click previous page etc.-it was drop down” (Participant 5) and all rated the Action Plan as being of appropriate duration. 

*Helpfulness:* All participants rated the ‘About AVP-IT’ page as helpful. Three out of four participants stated that the Action Plan Overview, How To, Tools, and Tips for Success was helpful, and at least four or five stated that the ‘My Toolkit’ page sections of About, How To, Tools, and Tips for Success were helpful. Fewer participants (two of five) perceived the ‘Testimonial’ section as extremely helpful or very helpful. 

Recommendations for improvement within the AVP-IT website included more role modelling stories on “challenges practices have had and how they overcame them” that are “tailored to the practice” (Participant 1) and to “show actual improvement in practices sites that had already used the website [using] examples of real-life scenarios” (Participant 2), and to show “Actual physician-family scenarios showing effective communication/actions taken by the provider to deal with specific parental concerns” (Participant 2). Recommendations for improvement of the tailored Action Plan included a comment on the challenge “that sometimes the person entering the answer may be unsure of the correct response. Since the survey only includes two responses options for some of the questions, it might be a little challenging” (Participant 4) and to “add information about anticipated response time” in the ‘Contact Us’ page (Participant 4). 

*Perceived impact*: All participants rated the information received in the tailored Action Plan as impactful in enabling clinics to adopt, implement, and maintain evidence-based strategies to increase HPV immunization. Participants stated that they “found it helpful,” (Participant 1), that “all practice sites can benefit from the information on the website,” (Participant 2), that “this can be an impactful tool … I think it is a great platform to get centers started in the right direction … [to] meet the needs of centers to enhance their vaccination strategy,” (Participant 4) and that “other clinics will see value in the tool … it gives a clear outline/road map to improve quality.” (Participant 3). One participant suggested the need for a partner to advertise at a statewide level (Participant 5).

*Motivational appeal:* All participants agreed that other clinics *should* use the AVP-IT website and will use the Action Plan. Most participants agreed that they would like to use the AVP-IT (4 of 5) but fewer agreed that other clinics *would* use the AVP-IT website (2 of 5). Participants indicated they found it appealing. Recommendations to improve AVP-IT user appeal included adding suggestions to add a section on “how to respond to specific parental concerns/comments” (Participant 3) in the tools page and to update the testimonials section to be more dynamic “since the photos were stagnant and did not change.” (Participant 4) 

*Relative advantage*: Participant ratings comparing the AVP-IT to usual practice (i.e., implementing strategies to increase HPV vaccination without the AVP-IT) were significantly skewed toward increased perceived ease and effectiveness (Table 9). 

#### 3.2.2. Step 6: Feasibility Case Study Results

Participating AVP Champions were medical assistants in the two participating clinics. They were female, Hispanic, between the ages of 20 and 29, reported their highest level of education as high school or GED (*n* = 1) and college degree (*n* = 1). At baseline, initiation of AVP strategies was classified as ‘pending.’ The exception was ‘parent education.’ This was classified as ‘partial’ because one clinic was providing parent education but had not yet provided the *HPVcancerFree* app. During the feasibility study, one of the champions resigned from the clinic to pursue other employment opportunities. This limited follow-up data to a single clinic site. At three-month follow-up, the strategies of AVP Champion, provider prompts, and parent education had been fully implemented and parent reminders had been partially implemented in accordance with AVP criteria (Table 10). Strategies of assessment and feedback and provider CME were ‘pending’.

### 3.3. Exit Interview Results

An exit interview was completed with one AVP Champion. She reported reviewing the AVP-IT Action Plan with her clinic manger and beginning to implement patient reminders by calling parents, since they did not have an automatic reminder system in place. She informed physicians to check if patients were due for the HPV vaccination through the state immunization tracking system (ImmTrac) and began referring parents to the *HPVcancerFree* app. She also reported that both the AVP continuing medical (CME) and nursing education (CNE) activity and AVP parent education app was being promoted through flyers hung up in the clinic as well as emails sent to clinic staff. Despite some challenges in gaining momentum on the AVP strategy rollout, she did not report any difficulty using the AVP-IT or her clinic’s tailored Action Plan, and she did not have any suggestions for improving the tool for future use.

### 3.4. Monitoring Calls

Although both champions completed the AVP-IT Action Wizard and obtained their tailored Action Plans without any trouble, they both encountered challenges with beginning the roll out of AVP strategies in their clinics. Little progress was reported during the first two months. The research team liaised with the deputy general manager of the clinic network who then provided closer oversight on the initiative. The order of rollout of AVP strategies was adjusted to commence with strategies that were less complex to implement (e.g., promoting and linking providers to the AVP-IT CME and promoting the *HPVcancerFree* parent education app).

## 4. Discussion

There is evidence for the efficacy of presumptive bundled messaging, assessment and feedback, provider prompts, patient reminders, and consumer education as strategies to increase HPV vaccination rates [19,20,21]. Historically, there has been uneven adoption and implementation of these strategies. Repeated in-person training and technical assistance have been conventionally employed to overcome individual and organizational barriers to implementing these strategies [47]. This study demonstrated that the evidence-based Adolescent Vaccination Program (AVP), a multi-component program originally delivered in-person in clinic settings to increase HPV vaccination, can be adapted to an online self-guided format. This online tool can provide guidance to implement these strategies, offering an innovative self-directed ‘do-it-yourself’ alternative to usual implementation practice.

The AVP-IT is designed to mitigate barriers to dissemination. Online delivery is intuitively appealing because it can offer intervention fidelity, tailor guidance to the clinic’s readiness to implement strategies, and reduce costs of training or materials. The utility of Internet-based decision support has been demonstrated in public health practice [28,29,30].

The premise for the adaptation of the AVP to the online AVP-IT (rather than development of the intervention from scratch) resided in the evidence for efficacy of the original AVP, the accessibility of the AVP creators to advise on adaptation, and the existence of original assets that could be modified for standalone use [37,38]. Irrespective, challenges in creating the AVP-IT included ensuring representation and fidelity to the original behavioral objectives, functionality of the original fully supported strategies, acceptability to end-users, feasibility for clinic-based delivery, and positive impact on implementation efforts. 

### 4.1. Usability

HCPs rated the AVP-IT as acceptable, credible, easy to use, simple, of sufficient scope, helpful, impactful, and appealing. Results of usability testing suggested that the AVP-IT could be acceptable as a standalone do-it-yourself intervention without the need for external support from a research or technical support team. This is consistent with the accruing evidence for the efficacy of digital decision support applications in impacting community health decision making [28,29,30]. Expert stakeholders were supportive of the design specifications to provide a simple, easy, rigorous (long digital lifespan), salient (tailored), and motivational intervention. The AVP-IT was also rated as adding significant thoroughness and fidelity to the implementation process. It was not, however, rated as a significant time saver. Feedback on the relative advantage of the AVP-IT was commensurate with those of similar clinic- and community-based implementation decision support [42,55]. The more temperate ratings for timeliness tend to occur in lockstep when the thoroughness and fidelity of a given practice are improved [40,42]. Usability is a necessary, but not sufficient, indicator of successful implementation in the clinic setting. Feasibility testing is an important next step in moving digital decision support interventions toward adoption. 

### 4.2. Feasibility

The feasibility testing case study also provides initial evidence for the feasibility of the AVP Implementation Tool (AVP-IT) in guiding clinic staff to implement the AVP unassisted. The AVP-IT is designed to provide high-fidelity stand-alone support that requires little-to-no intervention from developers outside the clinic. The AVP-IT remained accessible to clinic staff and the program functioned as designed in the field trial. No problems or disruptions to regular clinic functions were reported by participants in accessing and using the online AVP-IT website and creating and downloading the tailored Action Plan. During the feasibility test, the utility of the Action Plan was demonstrated with movement of three strategies (AVP Champion, provider prompts, and parent education) from ‘pending’ (not implemented) to ‘fully implemented’ by 3-month follow-up. Despite these changes, the feasibility testing highlighted a number of important facilitators and barriers to the successful deployment of the AVP-IT in clinics [32,45,46,47,49,50,52,53,54,56,57].

### 4.3. Facilitators and Barriers of Strategy Roll Out

The AVP-IT feasibility testing case study confirmed that individual- and clinic-level capacity was essential in successful Action Plan implementation of the AVP Champion, provider assessment and feedback, provider education, and patient education.

*AVP Champion*: Vaccination champions can encourage HPV vaccination as a clinic priority. In this feasibility study the role of the AVP Champion was to follow the AVP Action Plan guidelines for the roll out of AVP strategies. This role was prescriptive to provide clear and manageable tasks that could be transferred between staff members. Engagement in more elaborate promotional work, while potentially beneficial, was not expected. In this study, the clinic leadership recommended medical assistants (MAs) to be AVP Champions because these staff were based within a given clinic, familiar with clinic operations, and, unlike clinicians in this setting, had the operational bandwidth. The participation of MAs in this role was a departure from previous AVP studies and represented a ‘test case’ for this staff category. Even with the AVP-IT strategies to build champion capacity (tailored stepped Action Plan and a well-delineated website with informational videos), the MAs had difficulty accomplishing the tasks. This was due to interrelated issues of uncertainty about work priorities and competing demands to meet clinic metrics, expectations from leadership regarding this, and inertia to follow through on the Action Plan tasks. 

Leadership support was established at the front end of the study but insufficiently exploited during it. Providing a tailored Action Plan can be an innovative contribution to the field but is still subject to organizational barriers. This includes the concerns of clinic leadership regarding resource allocation to implement the recommended strategies that are dedicated to a single vaccination issue. Despite monthly updates by the research coordinator with the AVP Champions, there was initial inertia in enacting the Action Plan. While antithetical to the goal of a stand-alone implementation tool, clinic leadership was engaged by the AVP team to more actively direct the AVP-IT Champions. This catalyzed MA activity to implement the AVP strategies and underscores the importance of leadership involvement.

The study confirmed the advantages of having leadership or clinicians taking on the champions role, or, if a clinic lacked this capacity, to provide the appropriate support to the assigned champion. This study also reinforced the importance of champions having a solid understanding of the initiative as well as enthusiasm and the authority to drive the implementation process in the clinic setting. More testimonials featuring coping role models describing successful implementation (suggested in usability testing) provide important added training and motivation to champions, influencing normative perceptions as well as skills and self-efficacy. 

In previous studies, informational webinars had been used to train champions before each strategy rollout. Webinars were not employed within the concentrated timeframe of this feasibility study and may have played an important facilitation role. These are recommended for future work. The findings of this study highlighted the utility of checklists to accompany the Action Plan. These can provide a greater degree of ‘hand holding’ if lower-level staff are in the champion role. These also have facility in providing a concrete guide for leadership to assess progress and set common expectations. The feasibility of checklists is being assessed in a current field trial. 

AVP-IT field testing confirmed the importance of organizational factors of strong prescriptive leadership to support the implementation and help to prioritize competing demands on staff in busy clinics. This included regular status update meetings with clinic/network leadership, stepped written guides for staff at a more granular level than that of the Action Plan (e.g., checklists in accompaniment of monitoring calls), and enhanced tools to assist with rapid onboarding, orientation, and training to accommodate a continual staff turnover (e.g., training videos and testimonials of implementation success). Given this, it is premature to declare that AVP-IT has achieved ‘standalone’ status.

*Provider Assessment and Feedback:* The EHR system is critical to the HPV initiative because it provides tracking as well as prompting functions [54]. The quality of assessment and feedback reporting and evaluation success of the AVP-IT depends on the quality of data available within the system. The AVP has been successful in large pediatric clinic networks with robust data management systems. This study provided insight into the challenges of implementing the AVP in smaller safety-net clinic networks. The strategy to provide assessment and feedback reports to HCPs was not implemented (remained ‘pending’) in the current study due to incongruity between data elements required for the report and the data available within the clinic EHR. Modifications to the existing EHR was beyond the scope of this feasibility study, as was programming EHR-based provider prompts. To mitigate technological resource barriers, the specificity of assessment and feedback can be downgraded (e.g., reporting number of clinic encounters referencing HPV vaccination vs. identifying initiation and completion rates within the clinic population) until such time that clinics adopt greater breadth and specificity of data entry for HPV vaccinations. 

*Provider Continuing Education:* It is necessary that adequate knowledge and skills training be provided to HCPs and MAs who are frontline workers assigned to educate and motivate patients about the HPV vaccine [50]. Often, this training comes from in-person orientations and training (e.g., ACS, pharmaceutical representatives) [54]. Barriers to provider and staff trainings in busy clinics are that they involve provider absence that can adversely impact clinic productivity, provision of patient care, and revenue [54]. AVP-IT linkage to online CME was designed to provide scheduling flexibility to overcome this barrier while providing high fidelity skills training with ethics credit. Within the 3-month feasibility study, the AVP Champion had fully implemented the promotion about the CME. Immediate HCP registration in the CME was not observed. A longer-term implementation trial would allow sufficient time to determine if HCPs had incorporated the CME in their regular training schedules.

### 4.4. Parent Education

Patient-related barriers to HPV vaccination can include misinformation, language challenges, and low levels of literacy. Concerns of HCPs regarding antivaccine sentiments of stakeholders (parents) and perceived controversy over the stigma associated with the vaccine and the pervasion of internet-based antivaccine messaging have been reported as ‘outer setting’ barriers to implementation of HPC vaccination [58]. The AVP-IT provides adjunct materials such as posters and the AVP *HPVcancerFree* app (HPVCF) that highlight HPV vaccination, cuing parents to action and to help initiate conversation. The HPVCF app is designed to persuade parents by addressing perceived barriers. The impact of this has been previously described [25]. The AVP Champion fully implemented the promotion of the HPVCF. However, uptake of the app by parents was not observed. This study identified language barriers to existing materials that indicated the need for materials to accommodate non–English-speaking (Hispanic) patients. Addition of Spanish language materials and translation of the HPVCF is underway.

### 4.5. Limitations

Usability results should be interpreted in light of study limitations. Usability was conducted with five healthcare providers of varied job description. These respondents were sufficiently experienced and the sample size was appropriate for usability testing protocols that are designed to determine problems with the user experience, programming logic, and/or program bugs prior to field testing [43]. While statistical significance is less of an issue in prototype testing, conclusions regarding generalizability to other users, clinic types, and geographic region are naturally limited. 

Feasibility case study results also need to be interpreted in light of study limitations. Assessment of AVP-IT affect was restricted to immediate pre-post change in implementation within a limited 3-month time frame in two test sites (with loss of one clinic site). This is preliminary formative work. The results indicate the need for a more robust AVP-IT evaluation. Longer-term follow-up with a larger sample of clinics is necessary to evaluate the implementation and impact of the AVP-IT on HPV vaccination rates and provide an understanding of generalizability across clinics. 

A single-clinic experience is informative in the adaptation process but is in no way definitive of end results. Pilot feasibility studies are often limited in scope but sufficient to enable confidence that further field testing can be undertaken and to highlight anomalies such as program bugs, usability problems, or negative or harmful effects that indicate expanded testing may be premature. AVP-IT is being evaluated in a community-based trial in two safety-net clinic networks. More rigorous future RCTs are required to demonstrate the relative efficacy of this innovative digital approach compared with usual practice and could also inform on the cost-effectiveness implications of this implementation strategy. 

The feasibility trial highlighted an implicit paradox in implementation research, that a tool designed to facilitate implementation of an evidence-based program is, itself, subject to implementation barriers. The AVP-IT may facilitate the implementation of the AVP, but what facilitates the implementation of the AVP-IT? If the AVP-IT ‘works,’ how do you get clinics to use it? Does the implementation tool require an implementation intervention for its use? Do pre-implementation activities that build clinic capacity (e.g., EHR upgrades for assessment and feedback data assessment or patient reminders) constitute an implementation intervention for the implementation tool? 

### 4.6. Future/Policy

Progress in disseminating interventions to increase HPV vaccination has been pursued through academic institutions, government agencies, and private enterprises that have targeted clinic practices, provider knowledge and attitudes, and parent perceptions. Ongoing initiatives through vested national entities, including the National Cancer Institute (NCI), Centers of Disease Control and Prevention (CDC), and American Cancer Society (ACS); state organizations, such as the Cancer Prevention Research Initiative of Texas (CPRIT) and the HPV Coalition of Texas; and local organizations, such as the MD Anderson’s HPV Vaccination Initiative, have been contributing to mitigating the individual (attitudinal) and organizational (infrastructure) level barriers to HPV vaccination. These initiatives explore how to successfully facilitate the dissemination and implementation of evidence-based strategies to increase HPV vaccination. These initiatives may increase awareness and lower confusion regarding HPV vaccination guidelines, increase skills and self-efficacy for high-fidelity adaptation of strategies for the local clinic setting, and lower negative attitudes towards the HPV vaccination among HCPs. To date, the concerted efforts have demonstrated some success. HPV vaccination rates continue to rise with continued triangulation of efforts including increasing the vaccination eligible age, lowering dose requirements, inclusion of promotion of the vaccine among males, and shifts to presumptive bundled messaging. New innovations such as single-dose HPV vaccination are on the horizon. Contributory to this HPV-related ‘innovation agenda’ is the AVP-IT, which is among the first decision support tools that enable pediatric and community clinics to autonomously implement evidence-based strategies to increase HPV vaccination rates.

## 5. Conclusions

The AVP-IT and the provision of a stepped, tailored, clinic-based Action Plan that has acceptability as a guide to implement strategies to increase HPV vaccination may be feasible. Features of the adaptation, development, and testing of the AVP-IT reported here included a rigorous, stepped adaptation protocol with adaptation steps informed by iterative formative prototype testing (usability, feasibility, and perceived impact), providing empirical evidence to inform each subsequent development step. The AVP-IT decision support provides an Action Plan with tailored guidance to implement six evidence-based strategies (immunization champions, assessment and feedback, continuing education, provider prompts, parent reminders, and parent education). Healthcare providers rated the AVP-IT as acceptable, credible, easy, helpful, impactful, and appealing (≥80% agreement). They rated AVP-IT as making implementation easier and more effective compared to usual practice (*p* ≤ 0.05). Clinic-based AVP-IT use increased strategy implementation by 3-month follow-up. Study results need to be interpreted in the context of study limitations that include limited sample size, limited clinic-based testing, and no comparison sites. Irrespective, this study supports the translation of evidence-based programs for the clinic setting. The critical implementation facilitators and barriers described in the study can inform future work. The AVP-IT may provide an innovative contribution to accessible, utilitarian, and scalable decision support on implementing strategies to increase HPV vaccination rates in pediatric clinic settings. 

## Figures and Tables

**Figure 1 vaccines-11-01270-f001:**
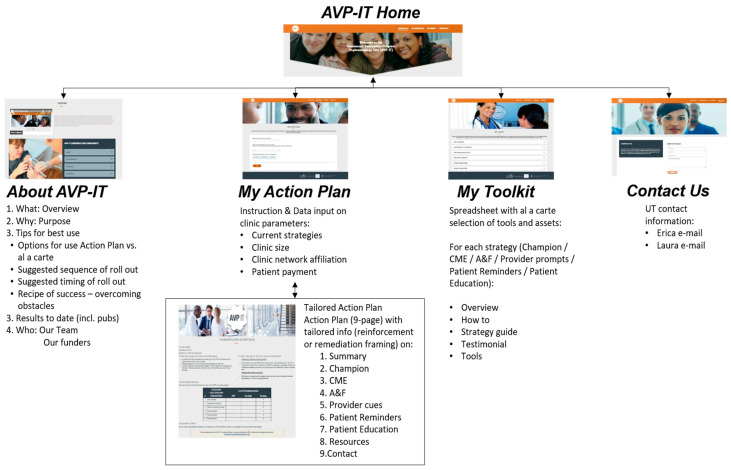
AVP-IT contents.

**Table 1 vaccines-11-01270-t001:** Phases and Steps in AVP Adaptation and Formative Evaluation.

Phase	Step	Description	Time
1. Adaptation	1	Semi-structured interviews in heterogenous clinics to establish facilitators and barriers to online immunization decision support.	July–October 2019
	2	Literature review on behavioral and environmental factors that impact adoption and implementation of evidence-based strategies to increase HPV vaccination and decision support interventions in clinic settings.	March–May 2019
	3	Design document describing the AVP-IT core content, scope, and methods and strategies.	June 2019–April 2020
	4	AVP-IT production.	May 2020–June 2021
2. Formative Evaluation	5	Usability Testing.	July 2021
	6	Feasibility Testing Case Study.	August–October 2021

**Table 2 vaccines-11-01270-t002:** Facilitators and Barriers to Implementation of Evidence-based Strategies: Provider Education, Provider Prompts, and Patient Reminders.

EBS ^1^	F/B ^2^	Issues
Provider Assessment and Feedback	F	Receiving printed copies of assessment and feedback reports [45]
Provider Education	F	Offering CME credit as an incentive to participate [46] Making a concerted effort to reach multiple providers per clinic [45] In-person delivery [45] Using content and materials from CDC (and like agencies) so practices understand they are receiving the most up-to-date information on best practices around HPV vaccination [47] Utilizing materials already developed and readily available to save time and help ensure consistent delivery of HPV-related messaging [47]

	B	Limited access to resources (e.g., multiple computers, conference space) for webinar trainings [45]
Provider Prompts	F	Making prompt location more visible [46] Adding prompt to the visit preparation materials [47] Discussing vaccination in the pre-visit huddle [47]
	B	Paper reminders interrupting workflow now that patient charts are electronic [34] Providers feel they already know what to do and do not need reminders [48] Reminder prompts not seen by providers [47] Reminder prompts ignored by providers [47,48] Reminder prompts contained missing/incorrect information [47,48] Lack of time to look at reminders [49,50] Paper reminders time-consuming to implement on busy days [47] Difficulty changing workflow and getting staff to regularly prompt providers [47] Alert fatigue [49] Limited time to track practice-specific vaccination rates or follow-up with patients [51,52]
Patient Reminders	F	Previous experience with the use of automated messaging systems and e-mails to patients [52]
	B	Structure of the automated reminder call decreased the amount of parents who opted-in for text message reminders [48] Non-automated reminders time-consuming to implement on busy days [47]

^1^ EBS = Evidence-based Strategy. ^2^ F = Facilitators, B = Barriers.

**Table 3 vaccines-11-01270-t003:** Facilitators and Barriers to Implementation of Evidence-based Strategies: Decision Support and Multi-component Programs.

EBS ^1^	F/B ^2^	Issues
Decision Support	F	“Right information, to right people, in right formats, through right channels, at right times to enhance decisions/outcomes” [32] Quality improvement and pay-for-performance initiatives [53]
	B	Lack of funding resources to employ a decision support system [50,53] Lack of training [49] Workflow barriers: busy practice schedules, incorporation of a new system, loss of productivity [49,51,53] Lack of buy-in/tool not useful [32,49] Inaccurate targeting leading to dissatisfaction and low compliance [32] Negative HCP and staff perceptions and attitudes towards technology adoption [32] Smaller practice size [53] Lack of confidentiality of patient information [53]
Multi-component programs	F	Frequent and consistent project reminders to staff [46] Identifying and engaging leadership teams for support [47] Training, technical assistance, and support from project staff [54] A designated provider champion to encourage program objectives, reinforce HPV vaccination as a priority, and to ensure that education and messaging consistently reaches all clinic personnel [54] Training to ensure that all team members had the knowledge necessary to implement the project [54] Accessible tools (e.g., posters highlighting HPV vaccination) to cue conversations with parents) [54] Formal processes to helped with implementation [54] Supportive leadership, especially when there are competing demands [54] Clinic staff actively communicating and promoting awareness among their peer network [54] Effective verbal communication among staff members [54]
	B	Staff turnover [45,46] Difficulties engaging leadership [45,46] System issues: difficulties with competing demands, limited staff time, stock of vaccine [45,46,48,54] Challenges in obtaining baseline HPV vaccination rates [54] Switching EHR systems [54] Trainings slow down productivity (e.g., interference of provision of care to patients [54] Lack of staff buy-in [54] Problems communicating to clinical and nonclinical staff with terminology and examples appropriate to their respective roles [54] Patient misinformation and vaccine stigma [54] Lack of resources to include non-English-speaking patients and patients with low health literacy [54] Lack of bidirectional communication between clinic EHRs and state immunization registries [54] Cost of program [54]

^1^ EBS = Evidence-based Strategy. ^2^ F = Facilitators, B = Barriers.

**Table 4 vaccines-11-01270-t004:** My Action Plan Wizard Data Input.

#	Evidence-Based Strategies		Response Options	Implementation Status		
				Pending	Partial	Full
1	AVP Champion	Is there an immunization champion in your clinic? (Y/N)	No	x		
			Yes		x	
		IF Yes: Does your champion devote time to HPV vaccination quality improvement? (Y/N)	Yes			x
2	Assessment and Feedback	In your clinic, are the HCPs given regular feedback (at least quarterly) on their HPV vaccination rates? (Y/N)	No	x		
			Yes		x	
		IF Yes: Does the feedback contain provider- and clinic-level data on all adolescent vaccinations, including comparisons between providers in your clinic and with national or clinic goals?’ (Y/N)	Yes			x
3	Continuing Education	In your clinic, are providers receiving continuing education on HPV vaccination? (Y/N)	No	x		
			Yes		x	
		IF Yes: Are your HCPs receiving CME or CNE with ethics credit that covers the latest training on HPV, HPV vaccination, evidence-based strategies, and best practices to navigate patient resistance? (Y/N)	Yes			x
4	Provider Prompts	In your clinic, do your HCPs receive prompts (in the EMR or otherwise) that a patient is eligible for the HPV vaccine? (Y/N)	No	x		
			Yes		x	
		IF Yes: Do the prompts identify patients who are both due and overdue for any dose of the HPV vaccine? (Y/N)	Yes			x
5	Parent Reminders	In your clinic, do you send reminders (e-mail/ text/ phone/ mail) to parents when their child is eligible for HPV vaccination? (Y/N)	No	x		
			Yes		x	
		IF Yes: Do the reminders identify patients who are both due and overdue for any dose of the HPV vaccine? (Y/N)	Yes			x

6	Parent Education	In your clinic, do you provide educational material on the HPV vaccine to your patients? (Y/N)	No	x		
			Yes		x	
		IF Yes: Do you provide any self-tailored phone-based Apps to raise their awareness about the importance and safety of HPV vaccination and also address myths and barriers surrounding HPV vaccination? (Y/N)	Yes			x

**Table 5 vaccines-11-01270-t005:** Summary of AVP Champion Tasks by Strategy.

#	Strategy	Description	Champion’s Tasks
1	Continuing Education (CE)	Online educational activity aimed to develop providers’ knowledge, skills, and performance in delivering HPV vaccination	Notify clinic staff about the AVP “Optimizing HPV Vaccination” online CE activity Note that the activity is an easy way to obtain ethics credit Ensure that all clinic staff (and new members) complete the CE activity
2	Assessment and Feedback	Reports that present providers with feedback on their performance in delivering adolescent vaccinations to their patients	Ensure availability of IT resources to perform data pulls on a quarterly basis Ensure that data is inserted into the assessment and feedback reports (this may be the Champion or an assigned clinic staff member) Print and distribute assessment and feedback reports to providers and clinic staff on a quarterly basis
3	Provider Prompts	Alerts that let healthcare providers know when patients are due for HPV vaccination	Work with your EMR vendor and learn if/how EMR-based provider prompts for HPV vaccination can be configured for your practice Ensure availability of IT resources to set up EMR provider reminders (or alternate reminders if EMR reminders are not possible) Train clinic staff on provider reminders
4	Parent Reminders	Reminders to let patients (parents) know when they are due for HPV vaccination	Work with your EMR vendor and learn if/how EMR-based parent reminders for HPV vaccination can be configured for your practice Ensure availability of IT resources to set up a parent reminder system Notify clinic staff about parent reminders
5	Parent Education	A self-administered app designed to raise awareness of HPV vaccination, reduce barriers to HPV vaccination, and enable parents to schedule HPV vaccination reminders through their smartphone	Notify clinic staff about the AVP *HPVcancerFree* app Train clinic staff to provide instructions to parents on how to download and use the *HPVcancerFree* app Ensure clinic staff provide the *HPVcancerFree* App to parents of all eligible patients Post flyers on the *HPVcancerFree* app in the clinic waiting room and exam rooms

**Table 6 vaccines-11-01270-t006:** Demographic Characteristics of AVP-IT Usability Sample (N = 5).

Characteristic	n
Gender	
Male	3
Female	2
Race/ethnicity	
Non-Hispanic White	4
Non-Hispanic black	0
Hispanic	1
Age	
40–49 years	1
50–59 years	3
60–69 years	1
Title	
Physician	3
Administrator	2

**Table 7 vaccines-11-01270-t007:** Acceptability: Usability Ratings for AVP-IT (N = 5) ^1^.

Construct	Item ^2^	# Agreement/ Total Respondents
Acceptability	*How much do you like or dislike the* …	Like a lot ^1^
	About AVP-IT page?	5/5
	Action Plan Wizard?	3/3
	Action Plan (by the 6 strategies)?	4/4
	Organization of each strategy by sections (Overview/How To/Tips/Tools)?	4/4
	Layout and format of the Action Plan?	4/4
Credibility	*I think the information I got from ….*	
	… the About AVP-IT page was accurate.	5/5
	… the About AVP-IT page can be trusted.	5/5
	… My Action Plan was accurate.	4/4
	… My Action Plan was trustworthy.	4/4
	… My Toolkit was trustworthy.	5/5
Ease of Use	*How easy or hard was it to* …	Easy ^1^
	… use the About AVP-IT page?	5/5
	… understand the About AVP-IT page?	5/5
	… complete the Action Plan Wizard?	3/3
	… use your Action Plan?	3/3
	… understand your Action Plan?	3/3
	… use the My Toolkit page?	5/5
		Agree ^3^
	I thought the AVP-IT was easy to use.	5/5
	I would imagine that most people would learn to use the AVP-IT very quickly.	4/5
	I felt very confident using the AVP-IT.	5/5
Complexity		Disagree
	I found the AVP-IT unnecessarily complex.	4/5 ^3^
	I think that I would need the support of a technical person to be able to use the AVP-IT.	4/5 ^4^
	I thought there was too much inconsistency in the AVP-IT.	4/5 ^3^
	I found the AVP-IT very cumbersome to use.	4/5 ^3^
	I needed to learn a lot of things before I could get going with the AVP-IT.	5/5
		Agree
	I found the various functions in the AVP-IT were well integrated.	5/5
Information Scope	*I think that the amount of information* ….	
	… on the About AVP-IT page was just right.	5/5
	… I got from My Action Plan was just right.	4/4
*Duration*		
	I think that the time it took to complete the Action Plan Wizard was just right.	4/4
Motivational Appeal	Do you think other clinics …	Yes ^5^
	should use the Action Plan?	5/5
	should use the AVP-IT website?	5/5
	will use the Action Plan?	4/5
	will use the AVP-IT website?	2/5
	I think that I would like to use the AVP-IT.	4/5

^1^ Ratings for those who used the feature (others responded ‘did not use’). ^2^ Questions stems are in italics ^3^ Other selected response/s was/were ‘neither agree nor disagree’. ^4^ Other selected response was ‘agree’. ^5^ Other selected response/s was/were ‘maybe’.

**Table 8 vaccines-11-01270-t008:** Utility: Usability Ratings for AVP-IT (N = 5) ^1^.

Construct	Item ^1^				# Agreement/ Total Respondents
Helpfulness	*I think the information I got from* …				
	… the About AVP-IT page was helpful.				5/5
	… My Toolkit was helpful.				5/5
	… My Action Plan was helpful.				4/4
	… My Toolkit was helpful.				5/5
	*For the Action Plan, how helpful do you think each section is* …	Extremely Helpful	Very Helpful	Somewhat Helpful	Not at all Helpful
	… Overview sections are for each strategy?	2/4	1/4	0/4	1/4
	… About sections are for each strategy?	2/4	0/4	2/4	0 (0)
	… How To sections are for each strategy?	2/4	1/5	0/4	1/4)
	… Tools sections are for each strategy?	2/4	1/4	0/4	1/4
	… Tips for Success sections are for each strategy?	2/4	1/4	0/4	1/4
	*For the Toolkit page, how helpful is each section in assisting you to implement strategies to increase HPV vaccination? …*	Extremely Helpful	Very Helpful	Somewhat Helpful	Not at all Helpful
	… About section?	3/5	1/5	1/5	0/5
	… How To section?	3/5	2/5	0/5	0/5
	… Tools sections?	3/5	2/5	0/5	0/5
	… Tips for Success section?	3/5	1/5	1/5	0/5
	… Testimonial section?	2/5	0/5	3/5	0/5
Perceived Impact	*I think the information I got from My Action Plan will help clinics …*			Strongly Agree	Agree
	… adopt strategies to increase HPV vaccination.			2/4	2/4
	… implement strategies to increase HPV vaccination.			2/4	2/4
	… maintain strategies to increase HPV vaccination.			3/4	1/4

^1^ Questions stems are in italics

**Table 9 vaccines-11-01270-t009:** Provider Ratings of AVP-IT (Comparison to Usual Practice) (*n* = 5).

Item *	Mean	SD	t	*p*
Ease of Use	1.40	0.55	6.51	0.003
Timeliness	1.80	1.30	2.06	0.108
Effectiveness	1.60	0.55	5.69	0.005

* Response set is a 5-point semantic differential scale ranging from 1–5 (is easier/is harder; takes less time/takes more time; is more effective/is less effective).

**Table 10 vaccines-11-01270-t010:** AVP-IT Feasibility Case Study: Change in Implementation of AVP Evidence-based Strategies from Baseline to 3-month Follow-up (*n* = 1).

#	HPV Vaccination Strategies	Level of Implementation of HPV Evidence-Based Strategies ^1^	
		Action Plan 1 Baseline	Action Plan 2 Follow-Up
1	AVP Champion	Pending	Fully
2	Assessment and feedback	Pending	Pending
3	Provider continuing education	Pending	Pending
4	Provider prompts	Pending	Fully
5	Parent reminders	Pending	Partially
6	Parent education	Partially	Fully

^1^ Pending = does not perform; partially = HPV EBS is being performed but not in accordance with the AVP Action Plan criteria; fully = implemented in accordance with AVP Action Plan. Refer to Table 4 for the definitions.

## Data Availability

The data presented in this study are available on request from the corresponding author. The data are not publicly available due to security restrictions.

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
