# Peer review of "Adaptation and Formative Evaluation of Online Decision Support to Implement Evidence-Based Strategies to Increase HPV Vaccination Rates in Pediatric Clinics"

_vaccines, 2023, doi:10.3390/vaccines11071270_

Round 1

Reviewer 1 Report

This is an interesting article and helps to develop strategies to increase HPV vaccination rate.

However,there are some shortcomings such as the small sample size of th survey, which will affect judgment of results.

In results 3.1.1, the author provide only subjective description without objective data.

Author Response

#

Reviewer Comments

Author response

1

This is an interesting article and helps to develop strategies to increase HPV vaccination rate.

We thank the reviewer for recognizing the contribution of this work.

2

However, there are some shortcomings such as the small sample size of the survey, which will affect judgment of results.

We agree that this study focuses on prototype development with modest preliminary formative usability and feasibility data. We have addressed this in relation to reviewer comments in Rows #10 and #21.

3

In results 3.1.1, the author provide only subjective description without objective data.

We have provided the conclusions of the qualitative enquiry by way of summary. The objective data would include transcripts and perhaps frequencies of the responses but we opted for a summary of the findings for brevity.

Reviewer 2 Report

The authors provide extensive information on the process of development and evaluation of an online decision support system that should help to implement evidence-based strategies to increase vaccination rates against human papilloma virus (HPV) in pediatric clinics. 

For readers who are not familiar with the US health care system the paper and its rationale may be somewhat difficultz to understand. Therefore, a series of changes in the manuscript are suggested.

A) Abstract:

please give details on geography (the whole project took place in the USA and results may not be as relevant to other geographical areas). Furthermore, it is unclear, when the study started and ended.

B) General Remarks:

the authors use abbreviations which are not necessarily self-explaining or well known to an international readership. Some are superfluous (e.g. Tdap and MCV in line 43 never appear again), some are related to the US health system (e.g. DS system, HCP, FQHCs, HMOs, EHR, A&F Reports). The reader finds some of the explanations of these abbreviations at various occasions (e.g., when checking Tables). It might be helpful to add at the beginnung of the paper a comprehensive list of the abbreviations used in the manuscript and the respective explanations.

Start/end/duration of the study are not clearly indicated in the text (only for the usability survey a time point is given, July 2021). Therefore, a graphical depiction of the different processes/steps shown in Table 1 in relation to a relevant timeline in years (X-axis) should be added as Figure 1 (instead of Table 1) - or the time frames should be added as a new column in Table 1.

Extensive use of of percentages in Tables 5 and 6 [n = 5]: I quote "As a general rule, when the numerator in count data is very small, or when the denominator is fairly small, I prefer to see the numerator and denominator separately rather than a percentage." [https://statmodeling.stat.columbia.edu/2010/06/24/sometimes_the_r/] . Please think about the data presented in Tables 5 and 6 (and the text) - 100% is quite strong a statement, but "5 out of 5" is equally strong and somehow more honest.

Feasibility testing using one (N=1) test person may be possible, but the reader asks himself or herself what information should be drawn from such a test. Table 8 suggests that something miore may be happening, but in the text the authors state that "one of the champions had resigned from the clinic..." - the feasibility study had been planned with N=2 test persons. The authors should explain, why they think, that 1 person is enough for fesibility testing.

Regarding Table 8 (and the decision process of the AVP-IT in general): after having finished reading the paper, the reader is not really able to understand how "Action Plans" are developed by the "AVP Champions". Please explain what these champions should do with AVP and which kind of help the online decision support (-IT) provides.

Minor points:

line 54: should read "patients aged"?

line 282: what is "an HPV Champion"?

Author Response

#

Reviewer Comments

Author response

1

The authors provide extensive information on the process of development and evaluation of an online decision support system that should help to implement evidence-based strategies to increase vaccination rates against human papilloma virus (HPV) in pediatric clinics. 

Thank you for this synopsis of our innovative online decision support approach.   

2

For readers who are not familiar with the US health care system the paper and its rationale may be somewhat difficult to understand. Therefore, a series of changes in the manuscript are suggested.

We thank the reviewer for consideration of international readers who may be unfamiliar with the U.S. health care system.

Abstract:

3

please give details on geography (the whole project took place in the USA and results may not be as relevant to other geographical areas). Furthermore, it is unclear, when the study started and ended.

We have edited the abstract to reflect national (U.S.) guidelines (Line 12), and location and time of the study (Line 15).

We have added a sentence to establish place and time for the study (Lines 104-105).

We have added a column to Table 1 to indicate timing of events (Table 1).

B) General Remarks:

4

the authors use abbreviations which are not necessarily self-explaining or well known to an international readership. Some are superfluous (e.g. Tdap and MCV in line 43 never appear again), some are related to the US health system (e.g. DS system, HCP, FQHCs, HMOs, I, A&F Reports). The reader finds some of the explanations of these abbreviations at various occasions (e.g., when checking Tables). It might be helpful to add at the beginning of the paper a comprehensive list of the abbreviations used in the manuscript and the respective explanations.

Thank you for this suggestion. We can see how this may provide difficulties for the reader. We decided not to introduce an abbreviations table but rather make edits in the text. We have attended to the following:

·   Tdap and MCV abbreviations appeared once and have been omitted (Line 43-44),

·   DS system – used only once and therefore expanded (Table 2),

·   EAG – Expert Advisory Group has been spelled out throughout.

·   FQHCs – spelled out (Line 320)

·   HMOs – spelled out (Line 325)

·   A&F Reports – A&F had been expanded (spelled out) throughout.

·   HCP – this is established in the abstract (Line 19) and text (Line 45). It is established again in Line 124. This term tends to be well recognized so we have left other instances as abbreviations.

·   CME – Line 208 provided abbreviation and left.

·   CNE – spelled out along with abbreviation (Line 418).

5

Start/end/duration of the study are not clearly indicated in the text (only for the usability survey a time point is given, July 2021). Therefore, a graphical depiction of the different processes/steps shown in Table 1 in relation to a relevant timeline in years (X-axis) should be added as Figure 1 (instead of Table 1) – or the time frames should be added as a new column in Table 1.

These are good suggestions. We have opted for the second option, to add the dates column to Table 1. Please also see Row #6 (above) in regard to edits.

6

Extensive use of percentages in Tables 5 and 6 [n = 5]: I quote “As a general rule, when the numerator in count data is very small, or when the denominator is fairly small, I prefer to see the numerator and denominator separately rather than a percentage.” [https://statmodeling.stat.columbia.edu/2010/06/24/sometimes_the_r/] . Please think about the data presented in Tables 5 and 6 (and the text) – 100% is quite strong a statement, but “5 out of 5” is equally strong and somehow more honest.

Thank you for this consideration. The sample size for usability studies is typically quite small as statistical significance is not required to make determinations of ‘success’ or ‘failure.’ We typically report percentages of agreement in these types of studies. However, given the reviewer’s concern in this regard, we have provided the respondent number as numerator and total respondent as denominator. Please see the results section on usability (section 3.2.1) and the associated Table (Table 6) in this regard. In the text if all participants were in agreement we just state that “all participants agreed / stated / rated…” and dispensed with percentages (e.g., 100%).

7

Feasibility testing using one (N=1) test person may be possible, but the reader asks himself or herself what information should be drawn from such a test. Table 8 suggests that something more may be happening, but in the text the authors state that “one of the champions had resigned from the clinic...” – the feasibility study had been planned with N=2 test persons. The authors should explain, why they think, that 1 person is enough for feasibility testing.

Thank you for expressing concern on this. We feel that the study describes a legitimate progression in formative work for a decision support prototype, but we are also concerned that we don’t ‘oversell’ the results given the constraints of the study. We opted to reframe the feasibility testing as a feasibility testing case study to temper perceptions on the nature of the study. We have modified text in this regard in Table 1 and Lines: 531, 543, 616, 630, 741, 744, 806. We also acknowledge the issue in the limitations section (Lines 748-752).

8

Regarding Table 8 (and the decision process of the AVP-IT in general): after having finished reading the paper, the reader is not really able to understand how “Action Plans” are developed by the “AVP Champions”. Please explain what these champions should do with AVP and which kind of help the online decision support (-IT) provides.

Thank you for alerting us to the need for clarity in this regard. We have provided a clearer delineation of steps in the use of the AVP-IT and referred to Tables 4 and 5 that provide specifics (Lines 362-383).

Note that due to modifications of the table structure (per Row #16) Table 8 is now Table 10.

Minor points:

9

line 54: should read “patients aged”?

Edit made (Line 55)

10

line 282: what is “an HPV Champion”?

We have edited this to read ‘AVP Champion’ (Line 289)

Reviewer 3 Report

HPV vaccination rates remain below national goals despite the availability of evidence ased strategies to increase rates. The Adolescent Vaccination Program (AVP) is a multi-component intervention demonstrated to increase HPV vaccination rates in pediatric clinics through the implementation of six evidence-based strategies. The purpose of this study was to adapt the AVP into an online decision support implementation tool (AVP-IT) for standalone use and to evaluate its feasibility for use in community clinics. Phase 1 (Adaptation) comprised clinic interviews (n=23), literature review, AVP-IT design documentation, and AVP-IT development. Phase 2 (Evaluation) comprised usability testing with healthcare providers (HCPs) (n=5) and feasibility testing in community-based clinics (n=2). AVP-IT decision support provides an Action Plan with tailored guidance on implementing six evidence-based strategies (immunization champions, assessment and feedback, continuing education, provider prompts, parent reminders, and parent education). HCPs rated the AVP-IT as acceptable, credible, easy, helpful, impactful and appealing (≥80% agreement). They rated AVP-IT supported implementation as easier and more effective compared to usual practice (p≤0.05). Clinic-based AVP-IT use facilitated strategy implementation by 3-month follow-up. AVP-IT promises accessible, utilitarian, and scalable decision support on strategies to increase HPV vaccination rates in pediatric clinic settings. Further feasibility and efficacy testing is indicated. 

Abbreviations are not allowed in the title.

All abbreviations must be defined (ex. of undefined one: AVP-IT and HPV).

Most of the tables are larger than the page

Which computational software was used to obtain the results, it should be indicated in the simulation section

References does not follow the journal style

Conclusions section should be provided with all the findings from the paper.

The statistical software used to process the data is not described

Is this enough from your point of view? "The sample comprised physicians (n=3), a clinic administrator (n=1), and a non-profit community manager (n=1)." Interpretation must be made.

In Table 7, the t-test was used, and it is not explained why and what the statistical hypotheses are used?

HPV vaccination rates remain below national goals despite the availability of evidence ased strategies to increase rates. The Adolescent Vaccination Program (AVP) is a multi-component intervention demonstrated to increase HPV vaccination rates in pediatric clinics through the implementation of six evidence-based strategies. The purpose of this study was to adapt the AVP into an online decision support implementation tool (AVP-IT) for standalone use and to evaluate its feasibility for use in community clinics. Phase 1 (Adaptation) comprised clinic interviews (n=23), literature review, AVP-IT design documentation, and AVP-IT development. Phase 2 (Evaluation) comprised usability testing with healthcare providers (HCPs) (n=5) and feasibility testing in community-based clinics (n=2). AVP-IT decision support provides an Action Plan with tailored guidance on implementing six evidence-based strategies (immunization champions, assessment and feedback, continuing education, provider prompts, parent reminders, and parent education). HCPs rated the AVP-IT as acceptable, credible, easy, helpful, impactful and appealing (≥80% agreement). They rated AVP-IT supported implementation as easier and more effective compared to usual practice (p≤0.05). Clinic-based AVP-IT use facilitated strategy implementation by 3-month follow-up. AVP-IT promises accessible, utilitarian, and scalable decision support on strategies to increase HPV vaccination rates in pediatric clinic settings. Further feasibility and efficacy testing is indicated. 

Abbreviations are not allowed in the title.

All abbreviations must be defined (ex. of undefined one: AVP-IT and HPV).

Most of the tables are larger than the page

Which computational software was used to obtain the results, it should be indicated in the simulation section

References does not follow the journal style

Conclusions section should be provided with all the findings from the paper.

The statistical software used to process the data is not described

Is this enough from your point of view? "The sample comprised physicians (n=3), a clinic administrator (n=1), and a non-profit community manager (n=1)." Interpretation must be made.

In Table 7, the t-test was used, and it is not explained why and what the statistical hypotheses are used?

Author Response

#

Reviewer Comments

Author response

1

Abbreviations are not allowed in the title.

Thank you for this notification. The abbreviation has been removed from the title.

2

All abbreviations must be defined (ex. of undefined one: AVP-IT and HPV).

Thank you for this suggestion. Please also see Row #7 comments in this regard. We have edited the text to rely less on abbreviations and to establish them at first use if they are used consistently. Specific to the examples mentioned here:

·   HPV has been provided spelled out in the abstract (Line 11) and HPV is spelled out in the Text (Line 33-34).

·   AVP-IT is defined in the abstract (Line 18) and in the text (Line 92), We decided to retain the abbreviation in the body of the text because it is ubiquitous to the subject of the study.

·   We have provided the full terms for other abbreviations that have less use (please refer to Row #7 above).

3

Most of the tables are larger than the page

We have cut the extended tables (and adjusted Table numbering and / or placed the tables such that they are on one page.

4

Which computational software was used to obtain the results, it should be indicated in the simulation section

Thank you for pointing this out. We have added reference to STATA-IC Version 17 analytic software (Lines 273).

5

References does not follow the journal style

Thank you for alerting us to this. We have revisited the references and checked citations. We noticed some edits were required for references mentioned in some tables and have edited accordingly.

6

Conclusions section should be provided with all the findings from the paper.

Thank you for this guidance. We have edited the conclusion to add study findings and have also reiterated the need to interpret findings in the light of study limitations.

7

The statistical software used to process the data is not described

Please see Row #17 above.

8

Is this enough from your point of view? "The sample comprised physicians (n=3), a clinic administrator (n=1), and a non-profit community manager (n=1)." Interpretation must be made.

Thank you for raising this. We did not address the sample size for the usability test as a limitation. This is because the sample is within a reasonable number for usability testing. Usability issues (UX, logic, function) can be assessed and mitigated before field testing with this type of sample which is typical for this work. We have moved reference to this to Lines 233-235 in the usability section. Given that this has been raised by the reviewer, we have also added a paragraph in the limitations section to address the small usability sample and draw caution to generalizability of results presented (Lines 734-740).

9

In Table 7, the t-test was used, and it is not explained why and what the statistical hypotheses are used?

Thank you for pointing this out. We have added text to address this:

‘The semantic differential scale was assessed using a paired 2-tailed t-test comparison of responses against neutral ratings using STATA analytic software. The hypotheses tested was that those exposed to the AVP-IT would perceive its use as providing significant advantage (easier, less time intensive, and more efficacious) compared to their usual practice.’ Lines 271-276.

Round 2

Reviewer 2 Report

Still percentages are used in Table 6 and 7 (1st line). The authors could skip them as well.

All the other points raised have been addressed, the paper reads well and is informative also to readers not very familiar with the US health care system.

Author Response

Thank you for these final comments. We have removed the percentages from Tables 6 and 7 (first line).

Reviewer 3 Report

The paper is an informed contribution to an adaptive and formative assessment of online decision support for implementing evidence-based strategies to increase HPV vaccination rates in pediatric clinics.

The paper is an informed contribution to an adaptive and formative assessment of online decision support for implementing evidence-based strategies to increase HPV vaccination rates in pediatric clinics. We accept the paper in its current form.

Author Response

Thank you for accepting the manuscript in its current form. We have also reviewed and edited the manuscript for readability, reducing compound and run on sentences. Edited paragraphs have been highlighted and are indicated by comments boxes reading "edited for readability.'